# LRRK2 Structure-Based Activation Mechanism and Pathogenesis

**DOI:** 10.3390/biom13040612

**Published:** 2023-03-28

**Authors:** Xiaojuan Zhang, Arjan Kortholt

**Affiliations:** 1Department of Cell Biochemistry, University of Groningen, Nijenborg 7, 9747 AG Groningen, The Netherlands; 2YETEM-Innovative Technologies Application and Research Centre, Suleyman Demirel University, Isparta 32260, Turkey

**Keywords:** Leucine-rich-repeat kinase 2, Parkinson’s disease, structures, activation mechanism, intra/intermolecular regulation

## Abstract

Mutations in the multidomain protein Leucine-rich-repeat kinase 2 (LRRK2) have been identified as a genetic risk factor for both sporadic and familial Parkinson’s disease (PD). LRRK2 has two enzymatic domains: a RocCOR tandem with GTPase activity and a kinase domain. In addition, LRRK2 has three N-terminal domains: ARM (Armadillo repeat), ANK (Ankyrin repeat), and LRR (Leucine-rich-repeat), and a C-terminal WD40 domain, all of which are involved in mediating protein–protein interactions (PPIs) and regulation of the LRRK2 catalytic core. The PD-related mutations have been found in nearly all LRRK2 domains, and most of them have increased kinase activity and/or decreased GTPase activity. The complex activation mechanism of LRRK2 includes at least intramolecular regulation, dimerization, and membrane recruitment. In this review, we highlight the recent developments in the structural characterization of LRRK2 and discuss these developments from the perspective of the LRRK2 activation mechanism, the pathological role of the PD mutants, and therapeutic targeting.

## 1. Introduction

Parkinson’s disease (PD), the second most common progressive neurodegenerative disease, affects more than 7 million people worldwide, and it is estimated that this number will be doubled by 2040 [1]. Besides environmental factors, in 10–15% of PD patients, genetic predisposition seems to play a major causative role. Through genome-wide association studies, about 20 PD-associated genes have been identified so far, including alpha-synuclein (*SNCA*), glucocerebrosidase (*GBA*), and Leucine-rich-repeat kinase 2 (*LRRK2*) [2,3]. *LRRK2* is a commonly mutated gene that emerges in both sporadic and familial PD [4]. Furthermore, recently, it was found that LRRK2 kinase activity was enhanced in idiopathic PD postmortem brain tissue [5].

LRRK2 is a large 286 KDa protein that comprises seven domains (Figure 1). The three N-terminal domains, ARM (Armadillo repeat), ANK (Ankyrin repeat), LRR (Leucine-rich-repeat), and the C-terminal WD40 domain are involved in mediating protein–protein interactions (PPIs) and regulation of the LRRK2 catalytic core. The catalytic core consists of a RocCOR tandem, Ras of complex protein (Roc) and C-terminal of Roc (COR), and a kinase domain. The RocCOR domain has GTPase activity and mediates LRRK2 dimerization, while the kinase domain can phosphorylate a subgroup of Rab proteins to control several cellular functions, including intracellular trafficking and autophagy [6,7,8,9]. A multitude of PD-related mutations have been identified over the different LRRK2 domains. The most common mutations, R1441G, Y1699C, and G2019S, are located within the catalytic core and show an increased kinase activity and reduced GTPase activity [10,11,12]. In addition, mutations in LRRK2 have also been linked to other diseases, such as Crohn’s disease (CD) and Hansen’s disease [13,14]. Interestingly, the R1398H variant, which is located within the Roc domain, has been reported to be protective in both PD and CD [15].

## 2. Molecular Structure of LRRK2

### 2.1. From Bacterial Roco to Human Full-Length LRRK2 Structures

Structural studies on LRRK2 have been ongoing for almost two decades. However, the expression of the separate LRRK2 domains and full-length protein has been a major challenge for a long time. So far, only the crystal structures of the human LRRK2 Roc and WD40 domain have been resolved [16,17]. In addition, X-ray structures are available from homologous Roco family proteins (Table 1). A major breakthrough in the field of structural biology in general and for determining the LRRK2 structure has been the recent developments in electron microscopy (EM). The possibilities related to stable and higher field emission EM, a robust vitrification apparatus, a volta phase plate, faster direct electron detector, and improved data analysing software have resulted in an increase in EM-based structure publications by over 70% in the last three years [18,19,20]. Using these techniques in combination with optimized purification protocols for full-length LRRK2 (flLRRK2) resulted in the first low-resolution negative-stain EM models of LRRK2 in 2016 [21,22]. More recently, the structures of a construct comprising Roc-COR-Kinase-WD40 domains (LRRK2_RCKW_) and flLRRK2 have been reported by Cryo-EM [23,24,25]. In addition, with Cryo-electron tomography (Cryo-ET), an in situ microtubule-bound full-length LRRK2 structure in cells was determined [26]. Another breakthrough in structural biology is the DeepMind-produced 3D protein structure prediction computational method, AlphaFold2, which makes it possible to predict reliable structures based on multiple sequence alignment and pair representation algorithms in silico [27,28].

In this review, we summarize the current structural knowledge of LRRK2 in the context of the complex activation mechanism of LRRK2 and discuss the potential mechanism of the disease-linked mutations.

### 2.2. Kinase Domain Structure

The LRRK2 kinase domain is considered the main output of the protein. It can phosphorylate a group of Rab GTPase family members to regulate several cellular pathways [33]. In addition, auto-phosphorylation of LRRK2 is also important for intramolecular regulation [34,35]. The protein kinase family was discovered by Krebs and Fisher during their study of glycogen phosphorylase in 1956 [36]. Kinase domains catalyze the transfer of γ-phosphate from ATP to a substrate. The kinase family is classified into three groups: Serine/Threonine kinase, tyrosine kinase, and atypical kinase. Based on the kinase domain sequence, LRRK2 belongs to the tyrosine kinase-like kinase (TKL) subfamily, a subfamily of Serine/Threonine kinase, whose members show sequence similarity to tyrosine kinases (TK) but lack TK-specific motifs [37,38].

A detailed structural map of the LRRK2 kinase domain first became available after the high-resolution LRRK2_RCKW_ and flLRRK2 Cryo-EM structures were published in 2019 [23,24]. The structures revealed that LRRK2 has a typical kinase structure with a rich β-sheets-formed N-lobe and an α-helix-composed C-lobe. The α-helix in the C-lobe forms the activation segment, which consists of the DFGψ (DYGψ in LRRK2) motif—activation loop—P + 1 loop—APE motif—α-F, and the N-lobe is tightly anchored to the kinase C-lobe by the β4-αC loop [38,39]. However, due to flexibility, the structure of the activation segment has only been partly resolved in the LRRK2_RCKW_ and flLRRK2 structures, while in the flLRRK2 structure, the DYGψ motif forms an unusual helix (Figure 2A). In the activation segment, the DYGψ motif works as a “brake” for LRRK2 conformational changes and activation: the Asp residue from the DFGψ motif binds with Mg^2+^ to indirectly interact with the oxygen of the β phosphate from ATP, and the P + 1 loop and APE motif provide the docking and interacting sites for substrates peptides, while the αF works as a scaffold that anchors the activation segment in a stable conformation in the kinase inactive state [40]. The two experimentally determined LRRK2 kinase structures both represent an inactive kinase conformation, with the activation segment collapsed on the protein interface in a locked position, thereby hindering substrate or ATP binding. In contrast, the predicted AlphaFold2 structure represents a DYG_out_ active kinase conformation (Figure 2B). The conserved GxGxxG motif in the N-lobe forms most of the ATP binding pocket and is crucial for the stabilization of the phosphate and catalytic activity. Interestingly, the GxGxxG motif adopts a different conformation in the LRRK2_RCKW_ and flLRRK2 (Figure 2A), which might be due to the different concentrations of ATP in the sample preparation.

By comparing the structures of the LRRK2 kinase domain with those of Braf (B-Raf Proto-Oncogene, Serine/Threonine kinase) and Src (Proto-oncogene tyrosine-protein kinase), the conserved Regulator spine (R-spine) and Catalytic spine (C-spine) were identified [41]. The C-spine has a substrate-capturing, nucleotide-binding, and phosphate transfer function, while the R-spine generally only assembles in kinase active state [42]. However, unexpectedly, the R-spine is also “visible” in the inactive LRRK2_RCKW_ and flLRRK2 structures. The highly conserved Phe in the DFGψ motif is one of the four R-spine residues, and LRRK2 has, instead of a DFGψ motif, an unusual DYGψ motif. Interestingly, mutating DYGψ to DFGψ resulted in an LRRK2 hyperactive phenotype, suggesting that Y2018 is important for stabilizing a kinase inactive conformation [43]. The DYGψ motif thus serves as a conformational switch that regulates LRRK2 activation.

### 2.3. RocCOR Structure

The LRRK2 RocCOR domain has GTPase hydrolysis activity and, in addition, regulates LRRK2 dimerization [44]. As discussed in more detail below, dimerization of LRRK2 is important for GTPase and kinase activation.

The first LRRK2 Roc crystal structure reported by Deng et al. in 2008 showed that the GDP-bound Roc domain has a typical G domain fold with five α-helix and six β-strands [16,45]. The Roc domain has a highly conserved P-loop (G1), switch I motif (G2), and switch II motif (G3) that together are responsible for Mg^2+^ and nucleotide binding. The G4-G5 motifs are important for guanine specificity, and are only partly conserved in LRRK2 [46,47,48]. The structure shows that two Roc proteins assemble in a dimeric structure by the interaction of one Roc N-terminal with the C-terminal of the other protomer, thereby forming a pseudo-twofold [16]. However, this unusual dimeric structure is not observed in structures of homologous Roco family proteins nor in any of the high-resolution LRRK2 structures. Furthermore, both an N and C terminal extension of the Roc domain can disrupt the dimerization. Therefore, this structure is most likely caused by a crystallization artifact that was induced by the short flanking boundaries of the construct used [49]. Biochemical and structural characterization of the RocCOR domain of *C. tepidum* (*Ct*) Roco has revealed instead that the dimerization is mainly mediated by the C-terminal of the COR domain (CORB) [29]. The full-length *Ct*Roco structure showed that in addition to the tight CORB interaction, the Roc domain is also involved in the dimeric interaction by Roc–Roc, Roc–CORA (N-terminal of the COR domain), and Roc–CORB interactions. These dimeric interfaces are mainly formed by the P-loop, switch II, and a highly conserved Roc dimerization loop between β4 and α3 [32]. An overlay of the *Ct*Roco structures and a monomeric *Mb*RocCORA of *Methanosarcina barkeri* (*Mb*) Roco2 structure show there are rearrangements of the P-loop, switch II, and CORA domain, suggesting that they might play an important role during the Roco G-protein cycle [31] (Figure 3A). Initially, it was proposed that Roco proteins form constitutive dimers; however, detailed biophysical and biochemical characterization of the *Ct*Roco proteins revealed a monomer–dimer transition during the G-protein cycle [29,50]. The protein is mainly monomeric in the GTP state, while it forms dimers in the nucleotide-free and GDP-bound states [50]. Furthermore, an analogue of LRRK2 PD-associated mutation stabilizes the dimer and shows decreased GTPase activity [50]. Together, this shows that dimerization is an important step in the Roco activation mechanism.

The LRRK2_RCKW_, flLRRK2, and predicted AlphaFold2 RocCOR structures all overlap well, with the exception of some differences observed for the Switch II in the LRRK2_RCKW_ structure (Figure 3B). All three structures show that the COR domain consists of two subdomains that are connected by a flexible linker [23,24]. CORA consists of multiple α-helices and a short three-stranded antiparallel sheet (Figure 3C). The CORB sub-domain consists of four flanked helices, a central seven-stranded antiparallel sheet, and a hairpin motif. Consistent with the bacterial Roco structures, the CORB domain is the main dimerization interface in the flLRRK2 structure, which is formed by two partially overlapped β sheets [24]. In addition, the LRRK2_RCKW_ single-particle and microtubule-associated structures revealed a WD40–WD40 domain interface [23,26]. How these COR–COR and WD40–WD40 interactions contribute to LRRK2 dimerization, and whether the nucleotide-dependent dimerization mechanism of *Ct*Roco is conserved in LRRK2, need to be determined.

### 2.4. N-Terminal and C-Terminal Scaffold Structure

The LRRK2 N-terminal ARM, ANK, LRR, and C-terminal WD40 domains are tandem repeat domains that most likely are involved in PPIs with upstream and downstream regulators [51,52,53]. Recent data suggest that, in addition, these domains have intramolecular interactions that are involved in regulating LRRK2 activity. Despite numerous efforts, structural information about the N-terminal domains remains scarce. Even in the recently published flLRRK2 structure, most of the ARM domain is still not resolved, probably due to its flexibility [24] (Figure 4). The ARM domain that is visible in the structure also does not show the expected typical three-helical structure [51]. The AlphaFold2 model suggests that the LRRK2 ARM domain groove is shallower than in other typical ARM structures, but it does predict 14 repeats in the region of aa49-702 that form a typical three-helix patched solenoid structure, with the shortest helix, H1, perpendicular to the left two antiparallel helices. The H1 and H2 are located at the cylindrical outer surface and are predicted to, together with H3, mediate most of the PPIs [54].

For the ANK domain, it was initially predicted that it consists of seven repeats from aa679-902, while a later study proposed it consists of five repeats from aa690 to aa860 [51,55]. However, both the flLRRK2 and AlphaFold2 structures show that the ANK domain only comprises three typical antiparallel helix repeats (aa705-795) that are connected by a hairpin (Figure 4). The LRRK2 ANK domain forms a typical L-shaped structure with the extended hairpin exposed to the outside to mediate PPIs [56,57,58]. The structures also revealed that the predicted C-terminal segment of the ANK domain (aa795-860) is actually part of the LRR domain, where it functions as a hinge helix that connects the LRR domain with the ANK, ARM, and WD40 domains [24].

The LRR domain of flLRRK2 is folded into a 14 + 1 repeat horse-shaped structure [24]. The beta-strand is located at the concave surface, and it is connected with the convex side α-helix by a loop [24]. After the last strand-helix repeat at the C-terminus of the LRR domain, there is a short linker that crosses the Roc–CORA linker and forms a connection with the Roc domain (Figure 4). In the *Ct*Roco protein, this linker between the LRR and Roc domain is around 40 residues in length and includes a PLxxPPPE motif that is conserved in prokaryotic Roco proteins [29]. This linker in *Ct*Roco was shown to be important for the stability of the Roc–COR construct protein, and the crystal structure showed direct interaction of the linker with the Roc domain switch II motif [29,32]. However, in both the flLRRK2 and AlphaFold2 structures, the linker and LRR domain are far away from the Roc catalytic core.

The crystal structure of the dimeric WD40 domain published by Zhang et al. in 2019 revealed a typical seven blade ring-like structure, each consisting of four anti-parallel β-strands [17] (Figure 4). The WD40–WD40 dimeric interface is formed by the circumference of blades V, IV, and VI, rather than by the popular blade top surfaces. Consistently, this same WD40 dimer interface is present in the microtubule-bound LRRK2 in situ structure and the trimeric LRRK2_RCKW_ EM structure [23,26]. Interestingly, the extended LRRK2_RCKW_, flLRRK2, and AlphaFold2 structures also show that the C-terminal helix of the WD40 domain directly interacts with the CORB and kinase domain, suggesting that the WD40 domain might regulate the enzymatic activity of LRRK2 [24] (Figure 4).

## 3. Structural Studies to Understand the LRRK2 Activation Mechanism

From biochemical and cellular data, it is clear that LRRK2 cycles between a monomeric cytosolic state that has low kinase activity and a high active dimeric membrane bound state [59] (Figure 5). In the cytosol, LRRK2 is most likely GTP-bound, and this monomeric conformation is stabilized by phosphorylation-dependent binding of 14-3-3 to LRRK2 s910 and s935 [60]. The flLRRK2 structures and cross-linking data show that in this inactive state, the WD40 domain is positioned very close to the N-terminal domains and even directly interacts with the N-terminus by the hinge helix (Figure 4). Schmidt et al. hypothesized that the WD40 domain together with the N-terminal domains works as the “brake” of LRRK2 activation by trapping LRRK2 in an inactive monomer state [41]. This hypothesis is also consistent with the active kinase structure and shifted N-terminal conformation in the AlphaFold2 model [41]. The binding of various Rab family proteins to the LRRK2 ARM domain can release this “brake” and induce membrane translocation [41,60,61]. The ARM domain extends to the outside of the LRRK2 dimeric core, far away from the other LRRK2 domains [21,22]. In this conformation, the ARM domain is accessible for binding to Rab proteins, as well as FAS-associated death domain protein (FADD) or with the E3 ubiquitin ligase C-terminus of Hsc 70-interacting protein (CHIP) to induce neuronal death or mediate LRRK2 degradation, respectively [62,63]. The high local concentration of LRRK2 at the membrane most likely induces dimerization of the protein. During dimerization, LRRK2-bound GTP is hydrolyzed and the kinase is fully active for substrate phosphorylation. The recently published Rab29-bound LRRK2 tetramer structure suggests that in addition to dimers, LRRK2 can also form oligomers at the membrane for activation [64]. Furthermore, the phosphorylated-membrane-localized Rab8 and Rab10 can bind to an additional site in the ARM domain, thereby facilitating a much stronger anchor for LRRK2, which results in prolonged LRRK2 activation [65] (Figure 5). From several studies, it is now clear that there is a correlation between GTPase and kinase activity; however, the exact mechanism of this interaction is still not completely understood [66,67]. Multiple studies have shown that Roc–CORA and the kinase domain are not in close proximity and thus cannot directly interact [21]. Instead, a recently published Hydrogen-Deuterium Exchange Mass Spectrometry (HDX-MS) study revealed that the Roc–CORA segment can induce a shift from a compact to an extended conformation during kinase activation [68]. Therefore, the interaction between the Roc and kinase domains may rely on moving the kinase domain to the Roc–CORA domain and simultaneously exposing the CORB domain to allow dimerization [68]. In addition, other domains could also be involved in the communication between the Roc and kinase domain. In the predicted AlphaFold2 model, and flLRRK2 compact and “J” shape structures, an N-terminal–Roc–CORA and a C-terminal CORB–kinase-WD40 axis are visible (Figure 4). The C-terminal CORB–kinase–WD40 axis is connected by a unique α-helix at the C-terminus, which is crucial for the stability of the LRRK2_RCKW_ protein [23]. The two axes cross at the kinase and ANK domain, while the kinase domain is embodied by the solenoid LRR convex side in both the LRRK2 compact and “J” shape structure [21,22,23,24]. This conformation may provide a suitable steric space for the kinase domain and the autophosphorylation clusters from the ANK–LRR and the LRR–Roc linker. It has been shown that autophosphorylation is important for both GTPase function and kinase activity [35].

Taken together, this shows that all LRRK2 domains are involved in the activation, phosphorylation, and/or regulation of the cellular localization. However, LRRK2 inactivation, monomerization, and dissociation from the membrane still need to be determined. In the presence of ATP-competitive inhibitors, LRRK2 accumulates in the form of well-ordered filaments on microtubules, where, in turn, it acts as a roadblock for both actin and dynein movement [23]. The mutations within the COR dimeric interface can significantly reduce the microtubule binding [25]. Moreover, a recent microtubule-associated LRRK2_RCKW_ structure revealed that the microtubule binding is mediated by two patches within the Roc domain [25]. However, the accumulation of LRRK2 on microtubules has only been shown in cells that overexpress LRRK2 [69]. Therefore, it is unclear if (and how) this plays a role in PD and targeting of LRRK2.

## 4. PD Mutation Localization and Potential Pathogenesis

The most common PD mutation, G2019S, is located within the kinase domain and has an increased kinase activity [12] (Figure 1). However, the structural analysis shows no difference between the conformation of the DYGψ motif in the G2019S and WT flLRRK2 structures [24]. Interestingly, the G2019S mutation generates an additional hydrogen bond with residue Q1918 in the αC-helix, which could stabilize the kinase N-lobe and C-lobe in a closed conformation to mediate the kinase activation [41]. This suggests that the G2019S mutation may modify the kinase activity by shifting the inherent active–inactive kinase state equilibrium. The I2020T most likely also stabilizes the kinase domain in an active state via an additional hydrogen bond [41,70]. This mechanism was confirmed by HDX-MS experiments that revealed WT LRRK2 kinase is predominantly in the inactive state, while PD mutations favor the active state [41].

The N2081D mutation is a risk factor for both CD and PD (Figure 1). This mutation has increased kinase activity and decreased acetylation of α-tubulin, which in cells results in an impaired lysosome response and autophagy [15]. Interestingly, N2081 is located in a helix in the kinase domain that is in close proximity to the LRR concave side, suggesting this mutation may alter the kinase activity by controlling the conformation of the N-terminal lid.

Several PD mutations have been identified in the RocCOR tandem domain (Figure 1). Most of these mutations are located in the Roc–COR interface and directly affect the GTPase activity and/or dimerization [68] (Figure 4). As described above, LRRK2 needs to go through a complete monomer–dimer cycle to become completely active. Thus, mutations shifting the equilibrium either to the monomer or dimer will change the LRRK2 activity. The R1441H mutation prolongs the LRRK2 active state and decreases GTPase activity by increasing the nucleotide affinity [49]. So far, there is no evidence that the R1441G/C mutations also affect the GTPase activity. However, it was shown that the R1441 mutations can disrupt the S1444 PKA consensus recognition site, and that 14-3-3 binds to S1444 in a phosphorylation-dependent manner [71]. The R1441 mutations could thus lead to decreased 14-3-3 binding, which, subsequently, can influence LRRK2 translocation and protein stability [72,73]. The LRRK2 Y1699C mutant strengthens the RocCOR intramolecular interaction and weakens LRRK2 dimerization, resulting in decreased GTPase activity [11]. The L487A mutation of CtRoco (analogous to LRRK2 I1371V) is located in the Roc–COR interface and induces stabilization of the dimer, thereby decreasing the GTPase activity [74]. The protective variant located in the RocCOR domain for both PD and CD, R1398H, also shows increased RocCOR dimerization and GTP hydrolysis [15,75].

The G2385R PD mutation is mainly found within the Chinese Han and Korean populations [17,76,77]. It is located in the WD40 domain and affects both the GTPase and kinase activity (Figure 1). Two potential mechanisms for G2385R have been proposed. The first hypothesis is that the G2385R disturbs the WD40 domain interface. The WD40 domain crystal structure indeed shows that G2385R mutation is located in the dimerization interface, and biochemical data suggest that this mutation disturbs LRRK2 dimerization [17]. However, in the flLRRK2 and AlphaFold2 structures, residue G2385 is not situated within the dimer interface, but instead faces the N-terminus. The second proposed mechanism states that the introduced positive charged Arginine disrupts the interaction with proteins, such as 14-3-3, Heat shock protein 90 (Hsp90), and Synaptic Vesicle (SV) proteins [78,79]. These impaired interactions could affect LRRK2 stability and localization, and thereby potentially lead to neurodegeneration. The same mechanism most likely also holds true for a less common WD40 domain-linked PD-mutation, G2294R, which affects the backbone of the WD40 β-propeller and reduces LRRK2 stability [80]. The CD-related mutation M2397T, which is localized at an inserted α-helix in the WD40 propeller blade, prolongs the LRRK2 half-life [81].

Many rare PD variants have been identified in the LRRK2 N-terminus (Figure 1), but the pathogenesis of these variants remains to be determined. However, mechanistically, these mutants most likely change the LRRK2 activation cycle by influencing PPIs. The E139K within the ARM domain could alter LRRK2 binding with Dynamin-related protein 1 (DRP1) and decrease LRRK2 binding with synuclein vesicles to promote vesicle fusion [82,83]. The E334K mutation could affect the ARM domain charge interaction for PPIs, and the mutation L550W could affect FADD binding to induce neuronal cell death [62]. There are also plenty of rare PD mutations in the LRR domain, such as I1006M, R1067Q, S1096C, Q1111H, I1122V, L1165P, I1192V, S1228T, and P1262A [84,85]. Mapping these mutations on the flLRRK2 structure shows that they are either at the convex or concave surface (Figure 4). Those mutations thus likely affect either intramolecular interactions (convex interfaces) or intermolecular interactions (concave surfaces) [86,87]. The only putative PD mutation in the ANK domain is R793M, which is located in the groove surface and close to the LRR-inserted hinge-helix, kinase activation loop, and the WD40 C-terminal helix (Figure 4).

Altogether, these studies show that, although almost all mutants result in increased kinase activity and reduced GTPase activity, the various mutations affect different steps in the activation mechanism.

## 5. Perspective

Recent developments in structural biology have provided key insights into the LRRK2 activation mechanism and have helped explain the mechanistic effect of important PD LRRK2 mutations. These structures also have revealed that various interfaces on LRRK2 are important for intermolecular and intramolecular interactions, dimerization, and activation. Importantly, these interfaces are potential targets to block LRRK2 activity [88,89]. There are still several important questions to be answered. Multiple studies have shown that LRRK2 cycles between an inactive cytosolic monomeric and active dimer/oligomeric membrane-bound state. However, so far, all high-resolution structures that have been solved are either from LRRK2 in solution or LRRK2 bound to microtubules. Furthermore, since the structure of the N-terminal domain has not been fully resolved, the binding mode of PPIs, and their role in membrane recruitment and intramolecular regulation, are not completely understood. It also remains to be determined which domains are involved in membrane binding and recruitment, how membrane-bound LRRK2 undergoes dimerization, how the cross-talk between the Roc and kinase domain is regulated, how the dynamic kinase conformation changes during substrate loading and phosphorylation, and how exactly the PD mutations affect these different steps in the activation mechanism. Detailed biochemical, biophysical, and structural analysis of LRRK2 interactions with PPIs and structures of the different activation states of LRRK2 bound to membranes will be instrumental for answering these questions.

## Figures and Tables

**Figure 1 biomolecules-13-00612-f001:**
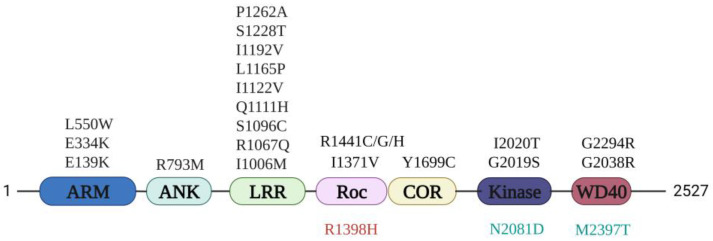
LRRK2 domain structure and Parkinson’s disease (PD) mutations. Armadillo (ARM), Ankyrin (ANK), Leucine-rich-repeat (LRR), Ras of complex proteins (Roc), C-terminal of Roc (COR), kinase (KIN), and WD40 domains. The PD-related mutations (black), protective mutation (red), and Crohn’s disease (CD) risk factor (green) are indicated.

**Figure 2 biomolecules-13-00612-f002:**
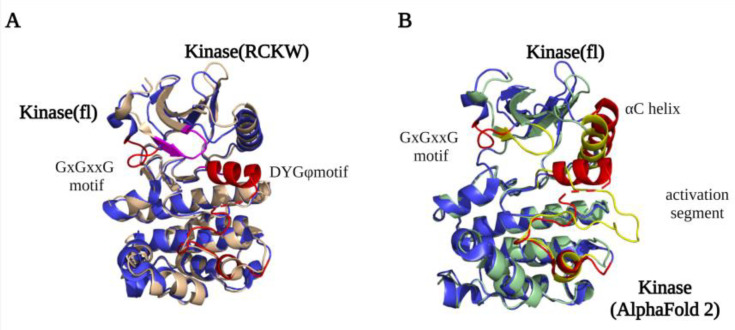
LRRK2 kinase domain structures. (**A**) An overlay of the kinase domain from LRRK2_RCKW_ (PDB ID: 6VP7) (light yellow) and flLRRK2 (PDB ID: 7LHW) (blue). The main difference between the structures is the conformation of the GxGxxG motif (highlighted LRRK2_RCKW_ in purple and flLRRK2 in red). (**B**) An overlay of the kinase domain from flLRRK2 (PDB ID: 7LHW) (blue) and AlphaFold2 model (green) structures. The GxGxxG motif, αC helix, and activation segments are highlighted in red (flLRRK2) and yellow (AlphaFold2), respectively.

**Figure 3 biomolecules-13-00612-f003:**
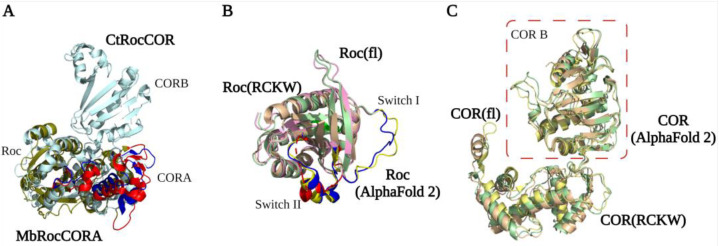
RocCOR tandem structures. (**A**) Alignment of the *Mb* (PDB ID: 4WNR) and *Ct* (PDB ID: 3DPU) RocCOR domains in dark green and light blue respectively. The rearrangement of P-loop, switch II, and CORA domain are indicated in red (*Mb*RocCORA) and blue (*Ct*RocCOR). (**B**,**C**) Alignment of the Roc/COR domain from LRRK2_RCKW_ (PDB ID: 6VP7) (light yellow), flLRRK2 (PDB ID: 7LHW) (pink), and AlphaFold2 (green). The partly and totally missed switch I/II in LRRK2_RCKW_ are highlighted in blue (flLRRK2), red (LRRK2_RCKW_), and yellow (AlphaFold2), respectively.

**Figure 4 biomolecules-13-00612-f004:**
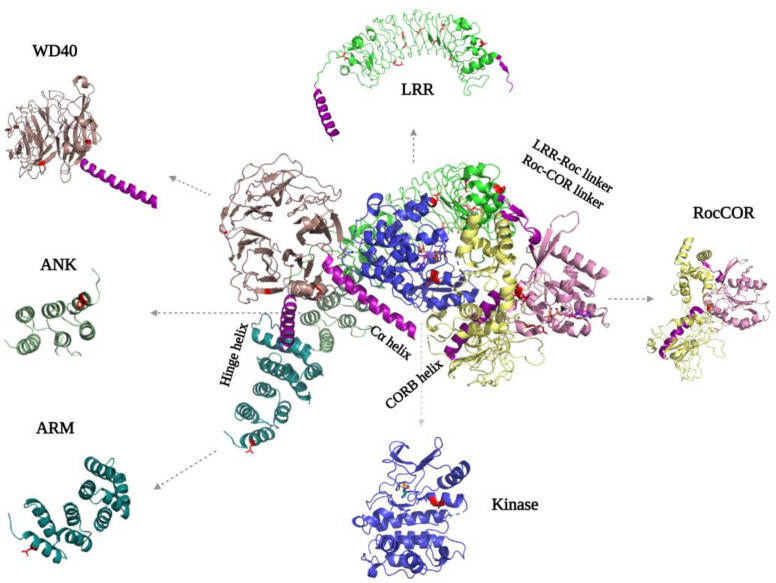
FlLRRK2 (PDB ID: 7LHW) with the separated domain’s structure on the sides. The helixes that are important for interdomain mediation are highlighted in purple. The PD mutations are indicated in both the full-length structure and separate domains in red, where they are located either in intradomain or interdomain interfaces.

**Figure 5 biomolecules-13-00612-f005:**
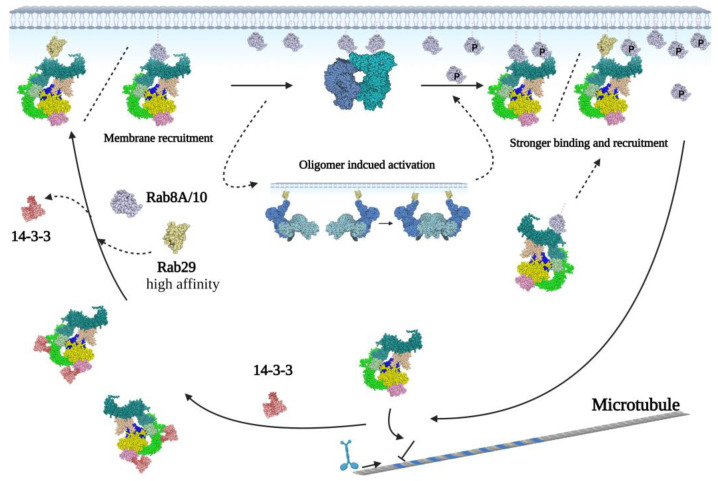
Proposed model of the LRRK2 activation cycle. LRRK2 is mainly monomeric in the cytosol, a conformation that is stabilized by binding to 14-3-3. Binding of Rab proteins to the ARM domain can induce recruitment of LRRK2 to membranes, where the high concentration of LRRK2 facilitates oligomerization and activation. Subsequently, the phosphorylated substrates, Rab8 and Rab10, can bind to the ARM domain to strengthen the LRRK2 membrane binding, thereby prolonging LRRK2 activation. (LRRK2 is indicated by AlphaFold2 model with: ARM in dark green, ANK in light green, LRR in green, Roc in pink, COR in yellow, kinase in blue, and WD40 in light yellow. The membrane-bounded dimeric/oligomeric protomers are indicated with blue or cyan).

**Table 1 biomolecules-13-00612-t001:** Published LRRK2 and Roco structures. *Hs* (*Homo sapiens*), *Ct* (*Chlorobium tepidum*), *Dd* (*Dictyostelium discoideum*), and *Mb* (*Methanosarcina barkeri*).

Year	Protein	Technology	Resolution	Expression host
2008 [16]	*Hs* LRRK2 Roc	X-ray crystallography	2.0 Å	*E. coli*
2008 [29]	*Ct* RocCOR	X-ray crystallography	2.9 Å	*E. coli*
2012 [30]	*Dd* Roco4 Kinase	X-ray crystallography	1.8 Å	*E. coli*
2015 [31]	*Mb* RocCORA	X-ray crystallography	2.9 Å	*E. coli*
2016 [21]	*Hs* LRRK2 fl	Negative stain EM	33 Å	HEK293F
2017 [22]	*Hs* LRRK2 fl	Cryo-EM	24.2 Å	HEK293FT
2018 [32]	*Ct* LRR-RocCOR	X-ray crystallography	3.29 Å	*E. coli*
2019 [17]	*Hs* LRRK2 WD40	X-ray crystallography	2.6 Å	Sf9 insect cell
2020 [23]	*Hs* LRRK2 RCKW	Cryo-EM	3.5 Å	Sf9 insect cell
2020 [26]	*Hs* LRRK2 fl	Cryo-ET	14 Å	HEK293T
2021 [24]	*Hs* LRRK2 fl	Cryo-EM	3.7 Å	HEK293F
2022 [25]	*Hs* LRRK2 RCKW	Cryo-EM	5.2 Å	Sf9 insect cell

## Data Availability

Not applicable.

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
