# Peer review of "LRRK2 Structure-Based Activation Mechanism and Pathogenesis"

_biomolecules, 2023, doi:10.3390/biom13040612_

Round 1
Reviewer 1 Report
This is largely a well written and informative article on LRRK2 structural biology from experts in the field. Some of the introductory genetics section could be improved and the manuscript needs editing for English language. The perspective section could be improved to provide more insight into remaining challenges from the authors viewpoint.
On line 27 of the intro there is a sentence with three words starting with capital letters. Actually this happens routinely throughout the paper. Generally a sentence is only started with a capital letter.
On line 29 of the intro, the genes would be better in capitals (lower case is generally used for mouse). Also the gene for GCase is GBA1.
“ Interestingly, unlike other monogenetic autosomal dominant PD mutations, LRRK2 is a high risk and commonly mutated gene that emerges in both sporadic and familial PD”. This is not a very precise or accurate statement. What is high risk and commonly mutated? Most LRRK2 mutations are very rare, some only in single families. Also genes are generally in italics. This needs to be checked throughout.
Figure 1 introduces a host of LRRK2 variants but it is not discussed in the text what these are. What is a PD-related mutation compared to a variant mutation, what is the protective mutation and what is CD? The section on LRRK2 genetics could be more accurately described. Why not include R1441H in figure 1 as it is mentioned later in the text?
For figure 2 I’m not sure wheat is a good colour choice/description. Yellow might be more understandable?
“CtRoco that have analogues PD-associated LRRK2 mutation showed increased dimerization and decreased GTPase activity.” This sentence doesn’t really make sense.
“The COR domain of LRRK2 is also folds into two subdomains”. This doesn’t really make sense.
“But the hairpin in the LRRK2 structures does not all extent outside of the helix”. This doesn’t really make sense.
“This mechanism was confirmed by HDX-MS experiments that revealed that WT LRRK2 Kinase is predominantly in the inactive state, while PD mutations favor the inactive state [65].” Is this correct?
On lines 324 and 326 G2385R is referred to only as G2385.
For experts in the field I found the perspective section light on. Can the authors provide more insight into specifically still needs to be done in this area and why?
Reviewer 2 Report
LRRK2 has emerged as a promising target for treating Parkinson's disease (PD) and is of significant therapeutic importance. This multi-domain protein possesses both GTPase and kinase activities, making it a complex and intriguing molecule. The current review paper covers recent structural discoveries related to LRRK2, explains its structure-based activation mechanism, and provides potential pathogenesis mechanisms for PD-linked mutations. Overall, this review is highly informative and provides a valuable understanding of the molecular mechanisms underlying PD, offering insights into potential therapeutic interventions.
However, there are a few areas that could benefit from improvement.
Firstly, the references could have been cited more accurately, which would enhance the credibility of the review.
1) This review missed an important paper published late last year in NSMB (Snead, D.M., et al. 2022), which produced a cryo-EM structure of the catalytic half of LRRK2 (bound to microtubules) and identified amino acids in LRRK2’s GTPase that mediate microtubule binding. Revisions should be made in Table 1 and Figure 5 accordingly.
2) Line 41: More references should be provided along with ref [5].
3) Line 51: Please provide references for all the listed disease-related mutations.
4) Line 67: Ref [16] is more appropriate to be moved to line 63.
5) The 5th reference in Table 1 is supposed to be ref [13] not [14].
6) Line 91: Please add references for the two structure papers published in 2020 and 2021, respectively.
7) Line 137: Ref [39] is the same paper as ref [9].
8) Line 181-184: Missing reference(s).
9) Line 214-216: Missing reference(s).
10) Line 236-238: Missing reference.
11) Line 254-256: Missing reference(s).
12) Line 260: Typo "622". Ref [62] is the same paper as ref [14].
13) Line 289-290: Missing reference(s).
Secondly, the presentation of the figures could have been clearer and more concise to better convey the information. For Figures 2, 3, and 4, I recommend that the authors use the same colors for the text labels as their corresponding structures. For the figure legends, I suggest adding "PDB ID" before each PDB ID in the text within brackets for clarification (e.g., flLRKK2 (PDB ID: 7LHW)). In Figure 3A, it is difficult to visualize the blue color, and in Figure 3C, it is hard to locate the purple color.
Other minor mistakes or typos are listed below:
1) Line 100: Missing "ψ" after "DFG."
2) Line 122: Missing "e" for "R-spin" and "C-spin."
3) Line 177-179: It is already known that COR-COR interaction is not the only dimerization interface in LRRK2. (Multiple studies have found WD40 dimerization.)
4) Line 298: Should be "while PD mutations favor the active state."
5) Line 299: "CD" was not defined earlier.
6) Line 315 and 317: These two statements about dimerization and GTPase activity conflict with each other.
Addressing these weaknesses would improve the overall quality of the paper.
